# Patient Specific Implants for Orbital Reconstruction in the Treatment of Silent Sinus Syndrome: Two Case Reports

**DOI:** 10.3390/jpm13040578

**Published:** 2023-03-25

**Authors:** Elisa Raveggi, Federica Sobrero, Giovanni Gerbino

**Affiliations:** Division of Maxillofacial Surgery, Surgical Science Department, Città della Salute e della Scienza Hospital, University of Turin, 10126 Torino, Italy

**Keywords:** enophthalmos, maxillary sinus, paranasal sinus diseases, orbit, patient specific implant

## Abstract

Silent sinus syndrome is a rare disorder characterized by ipsilateral enophthalmos and hypoglobus following a collapse of the orbital floor, in the presence of asymptomatic long-term maxillary sinusitis. It results in enophthalmos, hypoglobus and deepening of the superior palpebral sulcus. A standardized treatment protocol for this infrequent syndrome has not yet been established. The management includes restoration of maxillary sinus ventilation with functional endoscopic sinus surgery and orbital reconstruction, either concurrently or separately. In this paper, the authors presented two patients successfully treated with patient-specific implants, and intraoperative navigation. These cases highlight the benefit of computer-assisted planning and titanium patient-specific implants in the management of silent sinus syndrome. To the best of our knowledge, this is the first report that described the use of PSI with titanium spacers performed with the aid of intraoperative navigation for SSS treatment. Advantages, drawbacks of this technique and treatment alternatives currently available in the literature were also discussed.

## 1. Introduction

Silent sinus syndrome (SSS) is a rare disease characterized by a spontaneous collapse of the sinus walls and concomitant descent of the orbital floor, in the setting of a long-term asymptomatic maxillary sinusitis [1,2,3]. It was first reported by Montgomery at al. [4], who described it as chronic maxillary atelectasis resulting from a centripetal retraction of the sinus walls [1]. SSS, “chronic maxillary sinus actelectasia” (CMA) and “imploding antrum syndrome” were used as interchangeable terms by some authors, while others argued that CMA is a separate entity and SSS represents the last stage of CMA, when visual deformity and disturbances occur [2,5,6].

The aetiology of SSS is still unclear. Some authors suggested that SSS occurs in patients with a congenital hypoplastic maxillary sinus when an infection is acquired [7,8]. Others sustained that SSS is an acquired process initiated by an obstruction of the ostiomeatal complex, which may be post-surgical, post-traumatic or idiopathic [7]. The ostium obstruction could result in hypoventilation of the maxillary sinus and resorption of gas into the capillary system, which generates a negative pressure inside the sinus cavity [1,2,3]. This would cause subclinical inflammation of the mucosa lining, accumulation of mucus secretion, gradual weakening and erosion of the sinus bony walls and finally, maxillary atelectasia and wall collapse [1,2,6,7,9].

SSS is rare in children, while it commonly presents in the third to fifth decades of life, with no gender predilection [7,10,11,12]. It is usually monolateral [7,10,11,12].

Patients typically deny preexisting sinus disease or orbitofacial trauma [2]. Enophthalmos, hypoglobus and deepening of the superior palpebral sulcus are typical clinical features; they may be progressive over the course of several months or occur more rapidly. Other complaints such as diplopia and flattening of the malar region may be present [2,6,13]. Some authors considered the presence of sinonasal symptoms, e.g., rhinorrhoea, post-nasal drip, facial pressure, as an exclusion criterion for SSS [2].

The diagnosis is made of the basis of clinical and radiological findings. Computed tomography (CT) is regarded as mandatory for a reliable diagnosis of SSS [2,6,7]. Nevertheless, some studies suggested magnetic resonance (MR) imaging may be useful to evaluate dislocation of intra-orbital structures such as fat and extrinsic ocular muscles and rule out other pathology [2]. Differential diagnosis must consider other rare conditions such as facial hemiatrophy, progressive lipodystrophy and Horner syndrome, which may also produce enophthalmos [2,6,7].

The treatment goals are restoration of the maxillary sinus ventilation and correction of the aesthetic deformities (enophthalmos and hypoglobus) and diplopia, if present. The sequence of treatment, as well as the best technique for orbital floor reconstruction (OR), are a matter of debate. A variety of alloplastic and autogenous options were proposed for OR.

In this paper, two cases are presented where orbital reconstruction was performed with patient-specific implants (PSI) and intraoperative navigation, concomitantly or following functional endoscopic surgery (FESS). These cases highlight the advantages of computer-aided planning and patient-specific implants and introduce the novelty of being able to correct enophthalmos and hypoglobus intraoperatively with the aid of titanium spacers in patients with SSS. The authors also discuss the advantages and drawbacks of this technique and treatment alternatives currently available in the literature.

This report adhered to the principles of the Declaration of Helsinki and was written following the CARE-Case report Guideline. Patients gave written consent for the use of their data in this article.

## 2. Case Report 1

A 43-year-old female, in good general health, presented at the Division of Maxillofacial Surgery, Città della Salute e della Scienza Hospital, University of Torino (Italy), complaining of the sinking of the left eye and intermittent binocular diplopia since the previous month. She did not have a history of sinonasal symptoms, trauma, eye surgery or strabismus. Her clinical examination confirmed hypoglobus and enophthalmos of the left eye, deepening of the superior palpebral sulcus (Figure 1A) and slight pain on ocular movement.

A Hess–Lancaster test was performed, which revealed the presence of homonymous diplopia (Figure 2).

The remaining ophthalmologic examination was unremarkable. She performed a CT scan which revealed a collapse of the left medial wall and of the left orbital floor and a chronic sinusitis involving the left maxillary and frontal sinus and the left ethmoidal cells (Figure 1B).

Radiological findings and clinical features were compatible with the diagnosis of silent sinus syndrome. The surgeons opted for a one-step approach, performing the two procedures simultaneously, thus providing effective and timely correction of the aesthetic deformations.

Pre-operative CT data were uploaded into presurgical planning software (iPlan 3.0.5, BrainLab, Munich, Germany). The unaffected orbit was segmented as a virtual object and mirrored across the midline. The virtual plan was exported as an STL file and used to design a virtual titanium implant in collaboration with engineers (KLS-Martin, Jacksonville, FL, USA). The implant was designed in two complementary pieces to facilitate insertion. It included four trails along its surface which could facilitate a dynamic check of the implant position with the intra-operative navigation pointer (Figure 3A). In addition, three titanium spacers of different lengths (16, 18, 20 mm) and standard thickness (1.5 mm) were made in order to perform intra-operative clinical over-correction (Figure 3B).

After final approval, the PSI was fabricated. In the operating theatre, a left functional endoscopic sinus surgery (FESS) was performed, then the orbital floor implant was inserted through a transconjunctival approach and fixed to the orbital rim with five self-drilling screws. After positioning the PSI, a clinical evaluation was made: two titanium spacers were placed on the implant to obtain a clinically satisfying correction of hypoglobus, enophthalmos and superior palpebral sulcus. The navigation system allowed us to double-check the correct implant position and avoid the damage of important structures such as the optic nerve during the placement of the spacers.

A CT scan was performed the day after surgery, which confirmed correct implant and stable spacers position. The patient had an uneventful recovery and was followed for 18 months. After 6 months, the CT scan showed resolution of sinusitis and no displacement of the implants (Figure 1C).

The aesthetic result was satisfactory, with an acceptable correction of enophthalmos and hypoglobus (Figure 1D). The patient did not complain of diplopia in primary gaze, but a slight diplopia persisted in the upward and right lateral gaze, the extent of which, however, did not impact on diary activities.

## 3. Case Report 2

A 43-year-old female was referred with 8-month history of progressive left eye enophthalmos and diplopia in the extreme fields of gaze (Figure 4A).

Her medical history was silent except for an episode of decreased visual acuity and retroorbital headache after her second pregnancy. After the appearance of the enophthalmos, in June 2020, a magnetic resonance showed the presence of polypoid inflammatory tissue in the ethmoid-infundibular and fronto-nasal recess associated with concentric inflammatory thickening of the maxillary and frontal sinuses mucosa. In addition, a collapse of the medial wall and the floor of the left orbit were present. On the basis of the clinical history and radiologic findings, a diagnosis of SSS was made. The patient underwent a FESS of maxillary and ethmoidal sinuses in another hospital. Eight months after the endoscopic procedure, the visual deformity persisted. For this reason, the patient presented at the Division of Maxillofacial Surgery, Città della Salute e della Scienza Hospital, University of Torino (Italy). A clinical examination showed left eye enophthalmos and diplopia in the upward and lateral gaze, without decrease in visual acuity or muscular impairment. A CT scan demonstrated the restored pneumatization of the paranasal sinuses, while showing resorption and remodeling of the medial wall and floor of the left orbit, resulting in an increased volume compared with the contralateral side (Figure 4B).

A delayed reconstruction of the orbital floor was planned to restore the shape and volume of the orbit. In a similar fashion to the previous patient, virtual planning was used to obtain a mirrored image of the healthy orbit and a titanium PSI in two complementary pieces was fabricated. In addition, three titanium spacers were produced in order to eventually correct the orbital volume. The implant was inserted through a transconjunctival approach and fixed to the orbital rim with five self-drilling screws. The head surgeon evaluated the clinical correction of hypoglobus, enophthalmos and superior palpebral as satisfactory and, therefore, decided not to insert titanium spacers. Intraoperative navigation double-checked the correct positioning of the PSI (Figure 5).

The day after surgery, a CT scan was performed to check the PSI position (Figure 4C). Postoperative course was uneventful. The patient was followed-up for 12 months; correction of visual deformity and diplopia was satisfactorily achieved (Figure 4D).

Patient details of the two reported cases are summarized in Table 1.

## 4. Discussion

Silent sinus syndrome is a rare disorder characterized by ipsilateral enophthalmos and hypoglobus following a collapse of the orbital floor, in the presence of asymptomatic long-term maxillary sinusitis. A standardized treatment protocol for this infrequent syndrome has not yet been established [2].

In the management of SSS, different problems need to be addressed: maxillary sinus aeration and aesthetic and functional oculo-orbital issues deformities [3]. To restore normal maxillary sinus aeration, the ostial occlusion must be relieved [3]. At first, a Caldwell-Luc procedure was carried out [2]. Nowadays, the decompression of the sinus cavity is usually successfully achieved with endoscopic maxillary antrostomy and uncinectomy (FESS: functional endoscopic sinus surgery), which re-establishes the physiologic drainage pathway of the maxillary sinus [2].

The need and timing for performance of OR is still controversial [2,3]. Orbital floor reconstruction can be performed concurrently or separately from sinus surgery [2,3]. In a recent systematic review of 28 studies on SSS, Rosso et al. [2] reported that the majority of patients (116 out of 276 patients) underwent FESS alone, some were treated with FESS and OR either concomitantly or separately (72 patients), while a minority of patients underwent only OR (8 patients) or other management strategies. In fact, some authors performed endoscopic maxillary antrostomy alone, holding that it is a low-invasive procedure with satisfactory results [2]. Others preferred a two-stage approach, claiming that restoring the natural ventilation of the maxillary sinus may arrest or even reverse the orbital floor descent and improve enophthalmos, obviating the need for OR. For this reason, they delayed the decision for OR some months after antrostomy [2,14,15,16,17]. For example, Babar-Craig et al. [18] reported an improvement of enophthalmos in 14 of 16 patients over six months after sinus surgery, with only 2 patients requiring a secondary repair of the orbital floor. They reported mean improvement of enophthalmos of 2.2 mm (median 1.8 mm, range 0.5–4 mm). Similarly, Sivasubramaniam et al. [17] claimed that OR was unnecessary in 22 of 23 patients treated with FESS alone because of either complete or partial resolution of ocular symptoms. De Dorlodot et al. [14], in a small series on five patients, reported only one case in which OR was performed to correct a persistent visual deformity. In addition, Thomas et al. [15] suggested the two-stage approach would avoid the placement of an orbital floor implant in a potentially infected setting.

Finally, some authors advocated a one-step approach, performing the two procedures simultaneously, thus providing effective and timely correction of aesthetic deformations, reducing patient discomfort of a second hospitalization [2,16,19,20,21]. Cobb et al. [21] affirmed that while endoscopic antrostomy may halt the disease progression, “there is no reason to expect that retraction of the orbital floor and thus the position of the globe will be reversed”. Behbehani et al. [22] reported the simultaneous placement of orbital implants with the performance of FESS procedure with satisfactory outcome. They sustained that adverse events such as infection or visual loss are unlikely, while the single-stage approach obviates the need for a second hospitalization and general anaesthesia [22]. Arnon et al. reported [19] that a single-step procedure is a viable option which provides good aesthetic results and a quick rehabilitation. Similarly, Sesenna et al. [20] described three cases treated with simultaneous OR and restoration of maxillary sinus ventilation and concluded that this approach is advantageous in terms of reducing patient’s discomfort and hospitalization.

While Rosso et al. [2] suggested that the majority of SSS patients may be initially treated with FESS alone, reserving OR as a “rescue procedure” in case of persistent ocular disturbances, they admitted that consensus is lacking and “no study defined a quantitative cut-off which could help in the surgical decision” between a single or double-step approach. The authors, in agreement with other authors advocating a single-step procedure, believe that in the presence of a significant enophthalmos and/or diplopia, a concurrent OR is recommended [3,19,20]. In our series, both patients presented with a significant enophthalmos and functional problems. For the first patient, the authors were able to perform a combined FESS and OR. The second patient was referred 18 months after maxillary sinus surgery with persistent enophthalmos and diplopia, and so, a late OR was planned. In addition, the authors believe that the impact of the aesthetic and functional problems on the patient’s daily life should be taken in consideration when deciding to perform an OR. The presence of diplopia in primary gaze should be given careful consideration because of its important impact on quality of life.

In the secondary repair of acquired enophthalmos, the surgical approach, the type of material of OR and the necessity for volume overcorrection are still debated.

The surgical approach to the orbital floor and medial walls is a never-ending matter of debate in the literature. Depending on the anatomical structures which need to be visualized and reconstructed, different incision options are available. The transconjunctival and transcutaneous lower eyelid approaches are the most commonly used, while a transcaruncular or retrocaruncular extension can improve the exposure of the medial orbital wall [23,24]. For the management of OR in SSS, both the transconjunctival and the subciliary approaches were reported to be safe and appropriate [6,10,19,20,25,26]. The authors prefer a combined transconjunctival and transcaruncular approach, similarly to post-traumatic OR [27], because of its low invasiveness and good exposition of the anatomical structures. The authors believe that the transcaruncular extension is needed to correctly approach and visualize the medial orbital wall.

As for primary and secondary post-traumatic OR, various materials were proposed over the years to reconstruct premorbid bony anatomy. The options evolved from autogenous graft of bone or cartilage, to alloplastic materials (e.g., porous polyethylene or titanium), up to the most recent PSI [28,29]. Advocates of autogenous bone grafts hold that biocompatibility, lower risk of extrusion and foreign body reaction and infection are advantages of this option, but reabsorption and donor site morbidity remain a concern [3,7]. For example, Tieghi et al. [7] reported the use of an iliac bone graft covered with a Tutopatch sheet to avoid friction with the globe, resulting in uneventful healing and good aesthetic and functional recovery. Non-preformed and preformed orbital implants (e.g., titanium meshes or porous polyethylene implants) were used for decades in the repair of isolated single orbital wall fractures [29]. These type of flat implants were also reported for OR in SSS patients [30,31,32]. For example, Bovavolontà et al. [13] described the use of a Synpore mesh (combination of titanium and Medpor) in two patients, while Février et al. [32] reported the use of a titanium implant covered with an inert polymer (porous polyethylene, DePuy Synthes). Although the authors are experienced with the use of these type of implants for the reconstruction of orbital fractures, they believe that both non-preformed and preformed orbital implants are not adequate for OR in this setting. In SSS, the three-dimensional increase in orbital volume and the lack of anatomical landmarks due to bone remodelling make restoring the orbit with stock implants a real challenge, unlike what happens in the treatment of post-traumatic OR, when the orbital strut and the fracture ledges usually represent reliable anatomical landmarks for OR [3,29]. Bonavolontà et al. [13] and Arnon et al. [19] reported the use of virtual planning and navigation system to assist OR, while Baig et al. [3] were the first to describe the use of a PSI for OR in a patient with SSS. PSIs were employed with favorable results in different surgical subspecialties and anatomic locations. One of their varied applications is the reconstruction of orbital walls [33]. These personalized implants have been realized with different materials including porous polyethylene, and polyetheretherketone (PEEK), titanium, methylmethacrylate and hydroxyapatite [33,34,35,36,37]. Titanium PSIs have the advantage of being biocompatible, structural stable, radio-opaque, and ease to stabilize [25]. In addition, it is a technique that allows the surgeon to approach the surgery more easily. In a recent pilot study, Habib and Yoon [29] described the management of two patients with history of SSS and persistent diplopia and aesthetic asymmetry one year after functional endoscopic sinus surgery. A porous polyethylene PSI, thicker and wider than common preformed orbital implants, was positioned through a transconjunctival approach combined with a lateral canthotomy [29].

In the cases reported in this paper, the orbital defect involved both the medial wall and the orbital floor, so the OR needed a large PSI. To avoid the swinging eyelid approach, the authors decided to plan a titanium, two pieces complementary implant, customized to the patient’s defect, so that it could be easily inserted through a small surgical transconjunctival access. The radio-opaque titanium material had the advantage of allowing intraoperative navigation and postoperative CT control.

Prior studies showed that patients with late post-traumatic enophthalmos may require more than a pre-morbid reconstruction of the orbital walls to achieve adequate results [29,38,39]. The main mechanism of enophthalmos is an increase in bony orbital volume, with consequent change in shape and displacement of the orbital soft tissues [38,40]. However, additional mechanisms play a role, including orbital fat atrophy, fibrosis of orbital muscular and fat tissue and loss of ligament support [38,39,40]. A volumetric analysis revealed that the loss of orbital adipose tissue leads to an enlarged total orbital volume in patients with late post-traumatic enophthalmos, aggravating the deformity [29,38]. For this reason, in some cases, an OR only based on the mirrored unaffected orbit does not properly correct hypoglobus and enophthalmos [29,38,39]. There is no consensus in the literature on the method and quantity of overcorrection needed to achieve a satisfactory result. To further diminish the orbital volume, a bulkier PSI or additional implants may be used [26,39]. Directly designing a PSI with an overcorrection to the contralateral orbital floor is challenging because the amount of overcorrection needed is difficult to estimate, as orbital soft tissue behaves differently among treated individuals [26,39]. In addition, the volumes of the orbital cavities and the orbital soft tissue differ between the right and left side up to 8% [41,42]. To date, there is no consensus about the amount of overcorrection needed or an algorithm to calculate it from preoperative clinical or CT data [25,26]. To address this issue, the authors decided to use titanium spacers, previously described by Spalthoff et al. [26] and Singh et al. [25] for secondary orbital reconstruction. These spacers allow for a flexible intraoperative overcorrection of the orbital volume. In agreement with others, the authors believe that clinical judgment and experience are, to date, necessary to decide the site and degree of overcorrection in addition to the “true-to-original” bone reconstruction [25,26]. Future studies are needed to define an algorithm which may help the surgeon to predict the amount of overcorrection needed from available objective parameters.

Titanium spacers were positioned after PSI insertion in a retrobulbar position, until achieving a satisfactory clinical correction. The authors believe that superior palpebral sulcus depth is a reliable intraoperative clinical parameter to evaluate the quality of enophthalmos correction.

Dislocation of the spacers due to incorrect placement was reported in a few cases [25,26]. Spalthoff et al. [26] suggested that the addition of a fixation hole to the spacer and a rough undersurface may be useful to prevent this complication. In agreement with others [43], the authors hold that testing the implant movement on inferior rectus forced duction test is sufficient to establish whether it is in a stable position.

In accordance with previous reports [13,19], the authors strongly believe that computer-assisted navigated surgery has many advantages compared to conventional surgical techniques: it increases the accuracy of the PSI and spacers positioning, taking care not to damage important structures such as the optic nerve [44]. Despite the evident advantages of this technique, PSI are still expensive to produce, and computer-assisted navigated surgery is not yet a technology available in all centers [25].

## 5. Conclusions

In this paper, the authors presented two patients successfully treated with patient-specific implants, and intraoperative navigation. These cases highlight the benefit of computer-assisted planning and titanium patient-specific implants in the management of silent sinus syndrome. Clinical judgement and experience are still needed in the correction of severe late enophthalmos due to the lack of a reliable algorithm concerning the need and quantity of overcorrection eventually requested. To the best of our knowledge, this is the first report that described the use of PSI with titanium spacers performed with the aid of intraoperative navigation for SSS treatment. This approach might be another step forward in providing the best functional care and aesthetic outcome for patients with SSS.

## Figures and Tables

**Figure 1 jpm-13-00578-f001:**
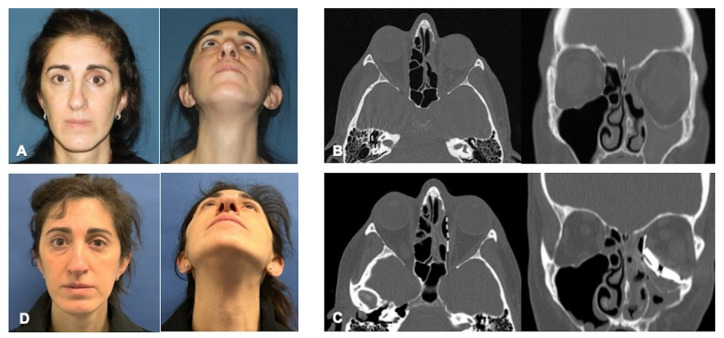
(**A**): Preoperative pictures at the time of patient presentation and diagnosis of left SSS, showing hypoglobus and enophthalmos of the left eye. (**B**): Pre-operative CT scan. Axial and coronal views showing completely opacified and atelectatic maxillary left sinus and collapse of the left orbital floor and medial wall. (**C**): Post-operative CT scan. Axial and coronal views showing correct implant placement, stable spacers position, and successful FESS surgery. (**D**): Clinical follow-up 18 months after surgery, showing satisfactory correction of the visual deformity.

**Figure 2 jpm-13-00578-f002:**
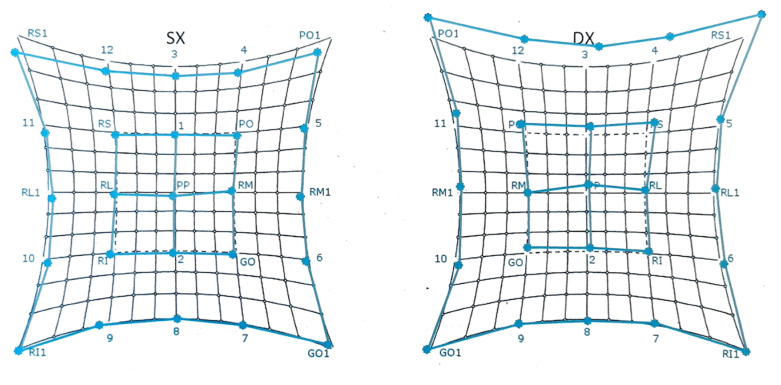
Pre-operative Hess–Lancaster test.

**Figure 3 jpm-13-00578-f003:**
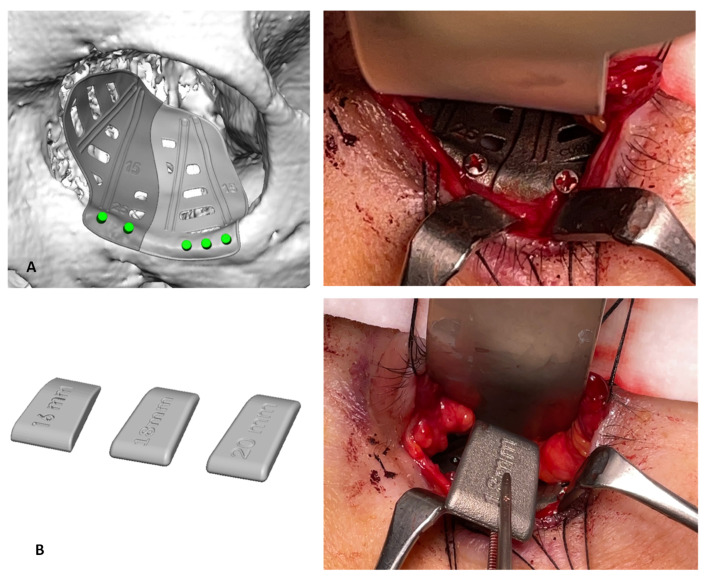
(**A**): The two complementary pieces implant: virtual design and intraoperative imaging. Green dots indicate the holes for screw placement; (**B**): titanium spacers of different lengths (16, 18, 20 mm) and standard thickness (1.5 mm): virtual design and intraoperative imaging.

**Figure 4 jpm-13-00578-f004:**
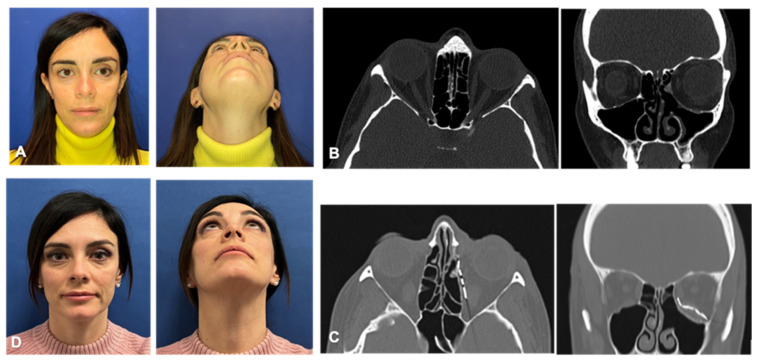
(**A**): Preoperative pictures showing left hypoglobus and enophthalmos. (**B**): Axial and coronal TC views demonstrating restored pneumatization of the paranasal sinuses, while showing resorption and remodeling of the medial wall and floor of the left orbit. (**C**): Axial and coronal post-operative CT views showing correct implant placement. (**D**): Clinical follow-up 12 months after surgery.

**Figure 5 jpm-13-00578-f005:**
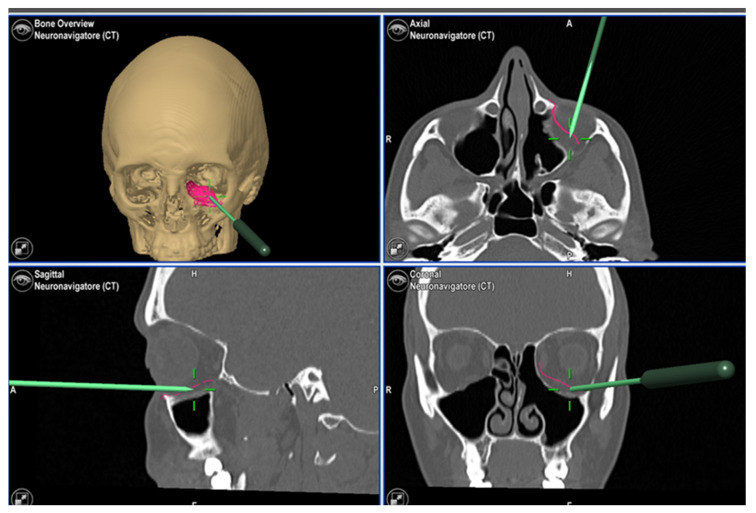
Screenshot obtained during intraoperative PSI navigation.

**Table 1 jpm-13-00578-t001:** Patients’ details and management.

Patient	Age	Sex	Diagnosis	Presentation	Surgical Timing	Implant Location	Implant Material	Titanium Spacers	Follow Up
1	43	F	Left silent sinus syndrome	Hypoglobus, enophthalmos, diplopia	Concomitant surgery	Medial wall and orbital floor	Titanium	Yes, 2	18 months
2	43	F	Left silent sinus syndrome	Hypoglobus, enophthalmos, diplopia	Two-stage approach	Medial wall and orbital floor	Titanium	No	12 months

## Data Availability

No new data were created. The manuscript is not under consideration by another journal, nor has it been previously published.

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
