# Peer review of "Patient Specific Implants for Orbital Reconstruction in the Treatment of Silent Sinus Syndrome: Two Case Reports"

_jpm, 2023, doi:10.3390/jpm13040578_

Round 1

Reviewer 1 Report

Thank you for your submission looking at the use of custom personalised implants for restoration of orbital volume in patietns with silent sinus syndrome. The results presented are good and you appear to have successfully corrected the dystopia and enophthalmos in both cases.

The use of custom printed implants, however, does not appear to add any benefit above and beyond the use of standard orbital reconstruction techniques, whether it be using bone or synthetic (titanium or Medpore, for example) implants. The aim of a custom implant is to accurately recreate the orbital shape and size, which it does, but then you are adding further small spacers to correct the position of the globe, which completely negates the use of custom implants. The custom implants are significantly more expensive that standard implant materials. 

Author Response

To the Editor,

We have thoroughly revised the manuscript, in particular the Discussion section, to meet the requirements, reaching the target number of words, references, and pictures/tables for the “Article” type of submission.

We thank the reviewers for their work and for the opportunity to improve our paper.

We tried to answer point-by-point to the comments of the reviewers.

# Reviewer 1.

Thank you for your submission looking at the use of custom personalised implants for restoration of orbital volume in patietns with silent sinus syndrome. The results presented are good and you appear to have successfully corrected the dystopia and enophthalmos in both cases.

 The use of custom printed implants, however, does not appear to add any benefit above and beyond the use of standard orbital reconstruction techniques, whether it be using bone or synthetic (titanium or Medpore, for example) implants. The aim of a custom implant is to accurately recreate the orbital shape and size, which it does, but then you are adding further small spacers to correct the position of the globe, which completely negates the use of custom implants. The custom implants are significantly more expensive that standard implant materials. 

Dear reviewer, first of all we would like to thank you for reading our paper and your comments/suggestions.

We modified the Discussion section to clarify you doubts and answer to your important opinion.

In accordance with other authors, we believe that PSIs add many benefits beyond the use of standard orbital reconstruction techniques. In fact, PSIs allow the surgeon to better deal with the problem of the three-dimensional increase in orbital volume and the lack of anatomical landmarks due to bone remodelling in patients with SSS, which make orbital reconstruction with stock implants a real challenge. Moreover, if the PSI is planned into two complementary pieces, it can be inserted more easily through less invasive approaches.

We agree that a PSI must accurately recreate the orbital shape and size. We think that orbital implants based on the mirroring of the healthy orbit represent the baseline for orbital reconstruction. Titanium spacers do not negate this function, because their aim is that of achieving a flexible intraoperative overcorrection. Prior studies have shown that patients with late enophthalmos may require more than a pre-morbid reconstruction of the orbital walls to achieve adequate results. However, to date, there is no consensus about the amount of overcorrection needed or an algorithm to calculate it from preoperative clinical or CT data. For these reasons, titanium spacers are a flexible solution which allows to achieve an overcorrection intraoperatively on the base of the clinical judgment.

We are aware that custom implants are expensive. At the end of the discussion the authors underlined that "Despite the evident advantages of this technique, PSI are still expensive to produce, and computer-assisted navigated surgery is not yet a technology available in all centers."

Dr. Elisa Raveggi MD,

Maxillofacial Unit, Surgical Science Departement

Città della Salute e della Scienza University Hospital

University of Torino, Torino, Italy

Reviewer 2 Report

The authors aimed to present two case reports of Silent Sinus Syndrome (SSS) treated through Patient-Specific Implants. This is a very interesting and revolutionary paper regarding the current treatment modality. However, there are some clarifications that the reviewer wants to make.

1. The title should carry the term "case report". For instance: "Case Report: Two Cases of Silent Sinus Syndrome Treated through Patient Specific Implants"

2. The last paragraph of the introduction should include how the paper adds to the current literature or its novelty. There are some studies on SSS that have not been mentioned. Here are some:

https://www.ncbi.nlm.nih.gov/pmc/articles/PMC8554165/

https://pubmed.ncbi.nlm.nih.gov/8746817/

https://pubmed.ncbi.nlm.nih.gov/28145930/

3. Still under introduction, the pathophysiology is present but the etiology may need to be included.

4. In the case presentation, Figures 1, 2, 4, and 5, as well as Figures 6, 7, 9, and 10, can be combined in a panel to improve the presentation. For instance: https://www.ncbi.nlm.nih.gov/pmc/articles/PMC8554165/figure/fig2/

5. Lines 211-212 claim that this is the first report of SSS treated with PSI. However, a study in 2021 reported two cases of SSS treated with PSI. Please include this in your discussion and compare/contrast the findings. https://pubmed.ncbi.nlm.nih.gov/34746511/

6. A table may be included to compare the two cases. For example: https://www.ncbi.nlm.nih.gov/pmc/articles/PMC8554165/table/tbl1/?report=objectonly

7. Mentioning the use of the CARE reporting guideline can also improve this manuscript. https://static1.squarespace.com/static/5db7b349364ff063a6c58ab8/t/5db7bf175f869e5812fd4293/1572323098501/CARE-checklist-English-2013.pdf

Author Response

To the Editor,

We have thoroughly revised the manuscript, in particular the Discussion section, to meet the requirements, reaching the target number of words, references, and pictures/tables for the “Article” type of submission.

We thank the reviewers for their work and for the opportunity to improve our paper.

We tried to answer point-by-point to the comments of the reviewers.

# Reviewer 2

The authors aimed to present two case reports of Silent Sinus Syndrome (SSS) treated through Patient-Specific Implants. This is a very interesting and revolutionary paper regarding the current treatment modality. However, there are some clarifications that the reviewer wants to make.

  1. The title should carry the term "case report". For instance: "Case Report: Two Cases of Silent Sinus Syndrome Treated through Patient Specific Implants"
  2. The last paragraph of the introduction should include how the paper adds to the current literature or its novelty. There are some studies on SSS that have not been mentioned. Here are some:
    1. https://www.ncbi.nlm.nih.gov/pmc/articles/PMC8554165/
    2. https://pubmed.ncbi.nlm.nih.gov/8746817/
    3. https://pubmed.ncbi.nlm.nih.gov/28145930/
  3. Still under introduction, the pathophysiology is present but the etiology may need to be included.
  4. In the case presentation, Figures 1, 2, 4, and 5, as well as Figures 6, 7, 9, and 10, can be combined in a panel to improve the presentation. For instance: https://www.ncbi.nlm.nih.gov/pmc/articles/PMC8554165/figure/fig2/
  5. Lines 211-212 claim that this is the first report of SSS treated with PSI. However, a study in 2021 reported two cases of SSS treated with PSI. Please include this in your discussion and compare/contrast the findings. https://pubmed.ncbi.nlm.nih.gov/34746511/
  6. A table may be included to compare the two cases. For example: https://www.ncbi.nlm.nih.gov/pmc/articles/PMC8554165/table/tbl1/?report=objectonly
  7. Mentioning the use of the CARE reporting guideline can also improve this manuscript. https://static1.squarespace.com/static/5db7b349364ff063a6c58ab8/t/5db7bf175f869e5812fd4293/1572323098501/CARE-checklist-English-2013.pdf

Dear reviewer, first of all we would like to thank you for reading our paper and your comments/suggestions. We modified the Discussion section to clarify you doubts and answer to your important opinion.

  1. We changed the title to: "PATIENT-SPECIFIC IMPLANTS FOR ORBITAL RECONSTRUCTION IN THE TREATMENT OF SILENT SINUS SYNDROME: TWO CASE REPORTS."
  2. Thank you for suggesting more interesting article. We added the citations in the manuscript.
  3. A paragraph on the etiology of SSS has been inserted in the introduction section.
  4. The figures have been grouped into panels as required.
  5. We have implemented the discussion section with the comparison with the case reports of Habib and Yoon. The differences compared to our work were:
    1. the type of material used for the PSI (Porous Polyethylene vs titanium)
    2. the surgical approach (swinging eyelid approach vs transconjunctival approach)
    3. the non-use of intraoperative navigation
    4. The non-use of titanium spacers.

For this, as reported in the manuscript, this is the first report that describes the use of PSI with titanium spacers for SSS treatment performed with the aid of intraoperative navigation.

  1. The table with the Patient Details has been inserted in the manuscript.
  2. CARE reporting guidelines have been mentioned.

We thank you again for your careful suggestions which allowed us to improve the scientific quality of our paper.

Dr. Elisa Raveggi MD,

Maxillofacial Unit, Surgical Science Departement

Città della Salute e della Scienza University Hospital

University of Torino, Torino, Italy

Reviewer 3 Report

This is an interesting and well-documented text describing a modern reconstructive technique. However, it has three comments.

Please explain and include in the discussion why it was chosen to add alloplastic blocks on an individual implant in the same surgical approach, rather than making a primary implant supporting the orbital contents at that level. When planning reconstruction with computer-assisted methods, there is flexibility of the implant shape created. And by applying the mirrorization technique, a high degree of accuracy can be achieved, for example, with regard to the planned hypercorrection in intraorbital soft tissue volume.

Considering such a two-part reconstruction presented by the authors [CAD/CAD + stock blocks], do you see any possibility of cost reduction in using a pre-bent titaniun mesh on a rapid prototyping model and then applying skock blocks to this mesh?

Also, please include in your Discussion or Introduction section an information on the choice of material from the available possible ones for craniofacial constrictions: bone from the cranial vault, ultra-high molecular weight polyethylene, zirconium oxide ect.

Overall, the manuscript shows great surgical sophistication, understanding of patient needs and knowledge of new technologies. The text is sure to meet the interest of readers.

Author Response

To the Editor,

We have thoroughly revised the manuscript, in particular the Discussion section, to meet the requirements, reaching the target number of words, references, and pictures/tables for the “Article” type of submission.

We thank the reviewers for their work and for the opportunity to improve our paper.

We tried to answer point-by-point to the comments of the reviewers.

# Reviewer 3

This is an interesting and well-documented text describing a modern reconstructive technique. However, it has three comments.

Please explain and include in the discussion why it was chosen to add alloplastic blocks on an individual implant in the same surgical approach, rather than making a primary implant supporting the orbital contents at that level. When planning reconstruction with computer-assisted methods, there is flexibility of the implant shape created. And by applying the mirrorization technique, a high degree of accuracy can be achieved, for example, with regard to the planned hypercorrection in intraorbital soft tissue volume.

Considering such a two-part reconstruction presented by the authors [CAD/CAD + stock blocks], do you see any possibility of cost reduction in using a pre-bent titanium mesh on a rapid prototyping model and then applying skock blocks to this mesh?

Also, please include in your Discussion or Introduction section an information on the choice of material from the available possible ones for craniofacial constrictions: bone from the cranial vault, ultra-high molecular weight polyethylene, zirconium oxide ect.

Overall, the manuscript shows great surgical sophistication, understanding of patient needs and knowledge of new technologies. The text is sure to meet the interest of readers.

Dear reviewer, first of all we would like to thank you for reading our paper and your comments/suggestions.

We modified the Discussion section to clarify you doubts and answer to your important opinion.

As we explained in the Discussion section, prior studies have shown that patients with late enophthalmos may require more than a pre-morbid reconstruction of the orbital walls to achieve adequate results and properly correct hypoglobus and enophthalmos. However, to date, there is no consensus about the amount of overcorrection needed or an algorithm to calculate it from preoperative clinical or CT data. For these reasons, titanium spacers are a flexible solution which allows to achieve an overcorrection intraoperatively on the base of the clinical judgment. Overcorrection may not be always necessary. Indeed, in case report 2, following the positioning of the PSI, the surgeon judged that adding any titanium spacer was not necessary.

Moreover, the use of titanium spacers means that we can use thinner PSI, which can be more easily inserted through less invasive approaches.

Your comment is very keen. We have adopted this solution in other cases of post-traumatic enophthalmos. This technique is a very good solution in these cases. However, in patients with SSS the the three-dimensional increase in orbital volume and the lack of anatomical landmarks due to bone remodeling make it difficult to use a pre-bent titanium mesh. PSIs a more precise and reliable option. Moreover, PSI can be planned in two complementary pieces, facilitating insertion through conservative approaches.

Information of the different materials for orbital reconstruction have been added in the Discussion section.

We thank you again for your careful suggestions which allowed us to improve the scientific quality of our paper.

Dr. Elisa Raveggi MD,

Maxillofacial Unit, Surgical Science Departement

Città della Salute e della Scienza University Hospital

University of Torino, Torino, Italy

Round 2

Reviewer 1 Report

Thank you for your response.

I agree that having a multi-part implant would make the surgery significantly easier and for this reason, PSI is of use in these cases. Thank you again for your submission, these can be challenging cases and your solution is a good one.

Reviewer 2 Report

The authors made substantial changes to their manuscript that improved the presentation of their study.